# A potentially abundant junctional RNA motif stabilized by m$^6$A and Mg$^{2+}$

Bei Liu[1], Dawn K. Merriman[2], Seung H. Choi[1], Maria A. Schumacher[1], Raphael Plangger[3], Christoph Kreutz[3], Stacy M. Horner [4,5], Kate D. Meyer[1] & Hashim M. Al-Hashimi [1,2]

$N^6$-Methyladenosine (m$^6$A) is an abundant post-transcriptional RNA modification that influences multiple aspects of gene expression. In addition to recruiting proteins, m$^6$A can modulate RNA function by destabilizing base pairing. Here, we show that when neighbored by a 5′ bulge, m$^6$A stabilizes m$^6$A–U base pairs, and global RNA structure by ~1 kcal mol$^{-1}$. The bulge most likely provides the flexibility needed to allow optimal stacking between the methyl group and 3′ neighbor through a conformation that is stabilized by Mg$^{2+}$. A bias toward this motif can help explain the global impact of methylation on RNA structure in transcriptome-wide studies. While m$^6$A embedded in duplex RNA is poorly recognized by the YTH domain reader protein and m$^6$A antibodies, both readily recognize m$^6$A in this newly identified motif. The results uncover potentially abundant and functional m$^6$A motifs that can modulate the epitranscriptomic structure landscape with important implications for the interpretation of transcriptome-wide data.

[1] Department of Biochemistry, Duke University School of Medicine, Durham, NC 27710, USA. [2] Department of Chemistry, Duke University, Durham, NC 27710, USA. [3] Institute of Organic Chemistry and Center for Molecular Biosciences Innsbruck (CMBI), University of Innsbruck, 6020 Innsbruck, Austria. [4] Department of Molecular Genetics and Microbiology, Duke University School of Medicine, Durham, NC 27710, USA. [5] Department of Medicine, Duke University School of Medicine, Durham, NC 27710, USA. Correspondence and requests for materials should be addressed to H.M.A.-H. (email: hashim.al.hashimi@duke.edu)

N[6]-methylated adenosine (m[6]A) is the most abundant internal RNA modification in eukaryotic mRNAs[1–3] and long non-coding RNAs (lncRNA)[4,5] and is also found in viral RNAs[6–8]. The modification is dynamically regulated by methyl transferase[9–11] and demethylase enzymes[12,13] that are linked to human disease[14,15]. It frequently occurs at the 3′ and 5′ untranslated regions of mRNA where it can influence RNA splicing[16,17], mRNA nuclear exportation[13,18], RNA decay[19], and translation initiation[20,21]. The modification is implicated in a growing number of processes including stem cell fate determination, stress response, DNA damage repair, microRNA biogenesis, lncRNA-mediated transcription repression, viral infection, and in the mechanisms of cancer[22–25].

m[6]A is thought to exert its biological effects through two general mechanisms. First, it can help recruit proteins that regulate mRNA fate by influencing splicing[16], export[26], decay[19,27], and translation initiation efficiency[20,27]. Many of these m[6]A reader proteins contain YTH domains, which specifically recognize the methyl group[5]. Second, m[6]A can exert biological effects by modulating RNA structure. In particular, studies have shown that m[6]A destabilizes A–U base pairing and RNA duplexes by 0.5–1.7 kcal mol[−1] [28,29]. This destabilization has been proposed to occur due to steric contacts between the N[6]-methyl group and the base[28] (Fig. 1a). Studies have shown that m[6]A can promote RNA melting and thereby enhance binding of single-stranded RNA binding proteins[30–32]. The modification can also disrupt non-canonical A–G base pairs (bps) required for protein binding[33]. Presence of m[6]A in mRNA also impedes tRNA accommodation within the ribosome and translation-elongation most likely by destabilizing A–U bps[34]. Interestingly, while

transcriptome-wide RNA structure mapping data indicate that m[6]A destabilizes RNA structure in vitro and in vivo[28,35], the destabilization is not observed at the m[6]A nucleotide per se, but rather, at the immediate 5′ neighbor which favors single-stranded conformations in methylated RNA[28]. The mechanism by which m[6]A destabilizes its 5′-neighbor is currently unknown.

Fewer studies have examined the potential for m[6]A to stabilize RNA even though there is precedent for this in the literature. Prior studies have shown that when placed at dangling ends of duplexes[28] or in apical loops[29], m[6]A stabilizes RNA by 0.1–1.2 kcal mol[−1] most likely due to favorable stacking interactions between the methyl group and adjacent bases. Here, by attempting to harness the destabilizing effects of m[6]A in the design of an RNA secondary structural switch, we discovered a bulge motif that is stabilized by m[6]A in a Mg[2+]-dependent manner. This motif is recognized by the YTH domain and m[6]A antibodies and can potentially help explain the global impact of m[6]A on RNA structure observed in transcriptome-wide studies.

## Results

**m[6]A stabilizes a junctional A–U base pair.** Our initial motivation was to examine whether the destabilizing effects of m[6]A on A–U base pairing could be harnessed to induce a change in RNA secondary structure. We designed an RNA hairpin (B5′, Fig. 1b) containing the most common[4,5] (GGm[6]ACU) m[6]A consensus sequence (DRACH, where D denotes A, G, or U; R is A or G; and H is A, C or U)[36–38]. This sequence is predicted[39] to fold into two iso-energetic secondary structures in which the m[6]A forms an A–U bp either at the junction next to a bulged guanine or deeper within the upper helix (Fig. 1b). The sequence is also predicted to

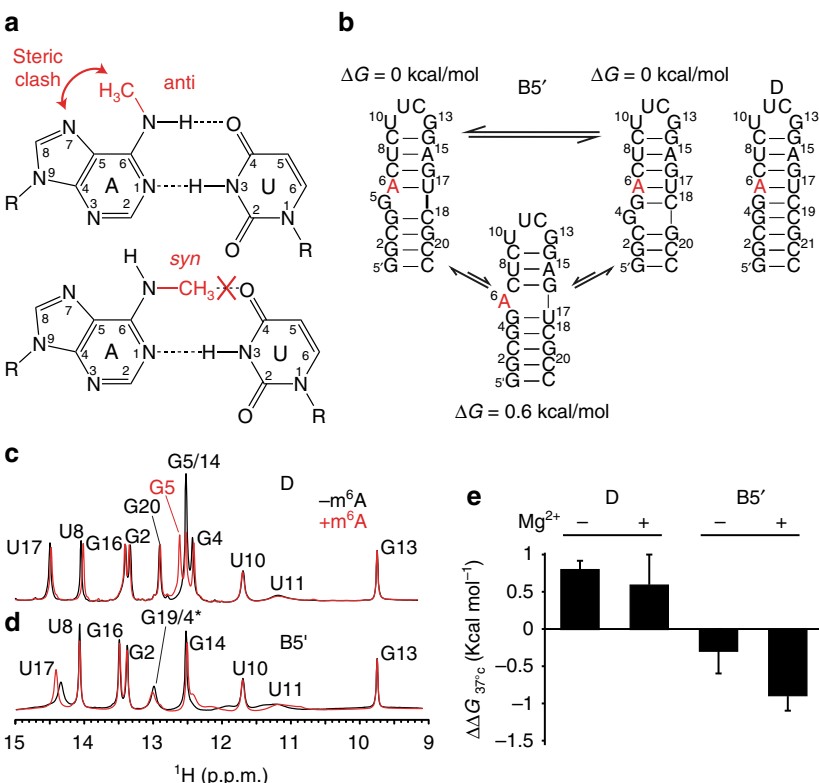

**Fig. 1** m[6]A stabilizes junctional A–U base pairs with 5′ bulge. **a** The methyl group in m[6]A destabilizes A–U pairing through steric contacts. **b** Design of an m[6]A-dependent RNA structural switch. Secondary structures and free energies computed using the MC-Fold web server[39]. The modified adenine (A6) is highlighted in red. **c**, **d** Comparison of [1]H NMR spectra of unmodified and m[6]A6 substituted (**c**) D and (**d**) B5′ recorded in 15 mM sodium phosphate, 25 mM NaCl, 0.1 mM EDTA and 3 mM Mg[2+] at pH 6.4 and 10 °C. **e** Impact of methylation on the thermal stability of the hairpins with and without 3 mM Mg[2+]. Shown are the differences in the free energy of melting between methylated and unmodified RNA, i.e., ΔΔG = ΔG(methylated) − ΔG(unmodified). The uncertainty reflects the standard deviation from at least three measurements (see Supplementary Table 1)

form an alternative conformation in which the m⁶A is bulged out and in which the 5′ neighboring guanine forms a non-canonical G–U wobble (Fig. 1b). This alternative conformation is predicted to be destabilized by 0.6 kcal mol⁻¹, which is within the range of m⁶A destabilization of A–U bps in duplex RNA[28,29]. Thus, we reasoned that m⁶A could destabilize the junctional A–U bp and promote formation of the alternative conformation containing the G–U mismatch, which can be readily detected by 1D ¹H NMR[40]. As a control, we also carried out experiments on a corresponding RNA duplex (denoted D), in which a cytosine nucleotide was included to pair with the bulged guanine (Fig. 1b). Experiments were carried out in the presence of 3 mM Mg²⁺.

The m⁶A substitution resulted in small perturbations in NMR spectra of the control duplex D (Fig. 1c). We observed the U17-H3 imino resonance of the m⁶A6 partner (Fig. 1c), which also has a nuclear Overhauser effect (NOE) cross peak with the amino group of m⁶A6 (Supplementary Fig. 1). This indicates that m⁶A6 forms a m⁶A6-U17 Watson-Crick bp stabilized by two H-bonds, as described previously for other RNA duplexes containing m⁶A[28]. Two NOE cross peaks with comparable intensities were observed between the m⁶A6 methyl proton and the base m⁶A6-H8 and m⁶A6-H2 (Supplementary Fig. 1). This indicates that the amino group either deviates from a perfectly *anti* conformation or rapidly (on the NMR chemical shift timescale) exchanges between the *anti* and *syn* conformations (see Supplementary Note).

The NMR spectra show that unmodified B5′ folds into a conformation with the predicted junctional A6-U17 Watson-Crick bp, bulged G5, and with G4 either forming a weak junctional bp or a bulge (Fig. 1d and Supplementary Fig. 2a, c). The m⁶A6 substitution did not lead to the predicted transition to the alternative conformation, as we did not observe the upfield shifted (10–12 ppm) imino resonances characteristic of G–U wobbles (Fig. 1d and Supplementary Fig. 2a). On the contrary, the substitution sharpened the U17 imino resonance, consistent with stabilization of the junctional Watson-Crick m⁶A6-U17 bp (Fig. 1d and Supplementary Fig. 2a). This together with uninterrupted NOE distance-based connectivity between U17 and G16 indicate that the methylation stabilizes the helical structure (Supplementary Fig. 3b). As in D, the NMR data indicate that while m⁶A6 forms a m⁶A6-U17 Watson-Crick bp, the methyl group deviates from a perfectly *anti* conformation (Supplementary Fig. 1 and see Supplementary Note). Therefore, counter to our predictions, and to the behavior observed in duplexes, m⁶A locally stabilizes the junctional A–U bp in B5′.

**m⁶A globally stabilizes B5′ in a Mg²⁺ dependent manner.** While the NMR data indicate that the m⁶A modification locally stabilizes the junctional A–U bp, this provides little information regarding the effect of the modification on overall RNA stability. We therefore used UV melting experiments to examine the impact of the m⁶A6 on RNA stability. Consistent with prior studies[28,29], m⁶A destabilized D by 0.6 ± 0.4 kcal mol⁻¹ at 37 °C. In stark contrast, m⁶A stabilized B5′ by 0.9 ± 0.2 kcal mol⁻¹ at the same temperature (Fig. 1e).

To further examine the basis for m⁶A-dependent stabilization, we carried out UV melting and NMR experiments in the absence of Mg²⁺. The modification destabilized D by a slightly greater amount (~0.8 kcal mol⁻¹) in the absence of Mg²⁺ (Fig. 1e). This was accompanied by broadening of the imino resonances in the absence of Mg²⁺ (Supplementary Fig. 3d), consistent with the destabilization observed using UV. In contrast, in the case of B5′, the degree of m⁶A stabilization decreased significantly from 0.9 ± 0.2 kcal mol⁻¹ to 0.3 ± 0.3 kcal mol⁻¹ in the absence of Mg²⁺ (Fig. 1e). NMR spectra, including data for site-specifically labeled

B5′ (Supplementary Fig. 2a), indicate that in the absence of Mg²⁺, unmodified B5′ folds into many distinct and slowly (on the NMR chemical shift timescale) exchanging conformations. The U17 imino resonance in methylated B5′ is significantly broadened, consistent with local melting of the m⁶A6-U17 bp in the absence of Mg²⁺ (Supplementary Fig. 2a and Fig. 3d). Here, the NOE data indicate that the methyl group adopts an *anti* conformation (Supplementary Fig. 1) and that G5 forms a G5-U17 mismatch (Supplementary Fig. 2a). Thus, the addition of Mg²⁺ induces a conformational switch relative to the structure without Mg²⁺ by stabilizing the junctional m⁶A6-U17 bp with a 5′ neighboring G5 bulge.

**Structural requirements for m⁶A stabilization.** Previous studies have shown that when placed at dangling ends, m⁶A stabilizes RNA duplexes by ~0.1–1.2 kcal mol⁻¹[28,29]. This has been attributed to favorable stacking and hydrophobic shielding of the methyl group. Interestingly, this stabilization is greater when m⁶A is placed at the 5′ (~ 1.2 kcal mol⁻¹) versus 3′ (~ 0.1–0.8 kcal mol⁻¹)[28,29] end of the duplex probably due to stronger stacking with the 3′ neighbor. Based on the thermodynamic parameters obtained from the UV experiments (Supplementary Table 1), the observed m⁶A-mediated stabilization of B5′ is driven by more favorable enthalpy, which is consistent with formation of favorable structural interactions. In particular, the 5′ neighboring guanine bulge could provide m⁶A enough "wiggle room" to form m⁶A–U WC bps in which m⁶A optimally stacks with the 3′ neighbor. This may require an unusual backbone conformation at the bulge, which is stabilized by Mg²⁺. Such conformations with optimal stacking may not be easily accommodated within the more rigid duplex interior, resulting in a net destabilization of the duplex.

To test the structural requirements for m⁶A stabilization, we varied the position and structural context of m⁶A (Fig. 2a) using three variants of the B5′ RNA (B5′_helical, B0, and B3′). We also examined a model RNA (HCV) derived from a naturally occurring m⁶A site identified in the 3′ UTR of the hepatitis C virus (HCV) genome[8]. In HCV, the m⁶A site is at the junction but on the strand opposite to the bulge. We performed NMR analysis in the presence of at 25 mM NaCl and 3 mM Mg²⁺ and confirmed that for all variants the methylated and unmodified constructs fold into their predicted secondary structures (Fig. 2b and Supplementary Fig. 3c).

Decreasing the conformational freedom available to m⁶A by moving the m⁶A–U bp one base pair deeper into the upper helix (B5′_helical, Fig. 2a) decreased the m⁶A dependent stabilization from 0.9 ± 0.2 kcal mol⁻¹ to 0.4 ± 0.4 kcal mol⁻¹ (Fig. 2c). Decreasing the conformational freedom by placing m⁶A opposite rather than immediately adjacent to the bulge in HCV resulted in a net destabilization by 0.4 ± 0.4 kcal mol⁻¹ (Fig. 2c). NMR spectra show that in both cases, the modification does not locally stabilize the m⁶A–U bp again possibly due to the lack of conformational freedom needed for optimal stacking (Fig. 2b).

Disrupting stacking with the 3′ neighbor through placement of a bulge 3′ to m⁶A in B3′ also resulted in a net destabilization by ~1.2 kcal mol⁻¹ (Fig. 2c). Interestingly, although m⁶A globally destabilizes B3′, it did locally stabilize the m⁶A6–U16 bp as evidenced by sharpening of the U16 imino resonance (Fig. 2b). Finally, placement of m⁶A in a bulge position destabilized B0 by ~0.7 ± 0.3 kcal mol⁻¹ (Fig. 2c). Here, the NMR spectra (Supplementary Fig. 1) indicate that the methyl group adopts a *syn* conformation, consistent with prior studies of single-stranded RNA[41], and that the modification induces the flipping out of the bulge adenine (Supplementary Fig. 4b). Loss of intra-helical stacking with the adenine bulge could explain the m⁶A-induced

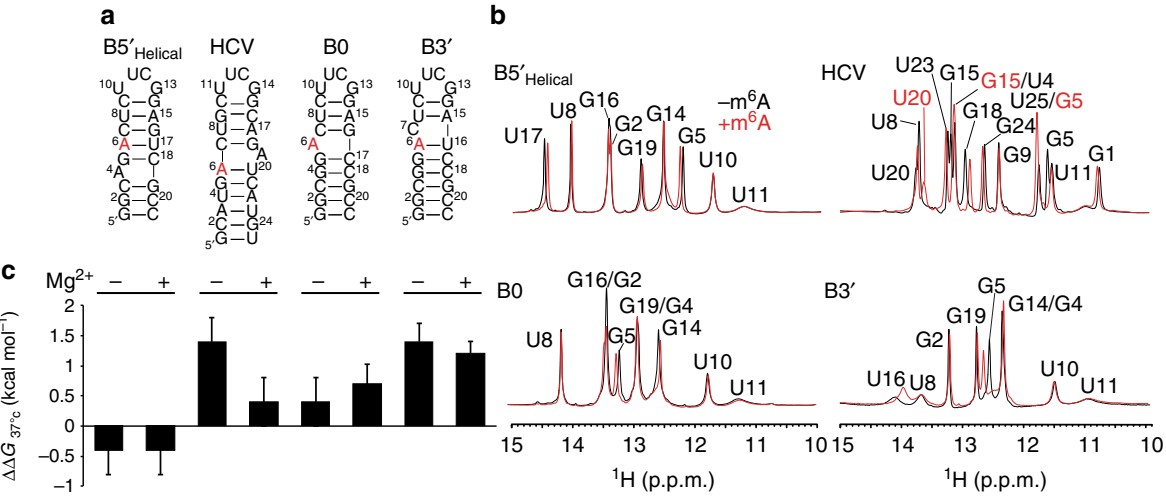

**Fig. 2** Structural requirements for m6A stabilization of junctional A–U base pairs. **a** Hairpin constructs that vary the secondary structural context of m6A. Secondary structures are predicted using MC-fold[39] and supported by NMR data (see Supplementary Fig. 3c). The HCV sequence is derived from the base of stem loop 1 (SL1) in the 3′ end of HCV genome (nt 9633-9642 linked by UUCG to nt 9668-9678) from the genotype 2A JFH1 strain (GeneBank accession: AB047639 [https://www.ncbi.nlm.nih.gov/nuccore/AB047639]). **b** Comparison of $^1$H NMR spectra of unmodified and m6A6 substituted hairpin constructs recorded in 15 mM sodium phosphate, 25 mM NaCl, 0.1 mM EDTA and 3 mM $Mg^{2+}$ at pH 6.4 and 10 °C. **c** Impact of methylation on the thermal stability of the hairpins with and without 3 mM $Mg^{2+}$. Shown are the differences in the free energy of melting between methylated and unmodified RNA, i.e., $\Delta\Delta G = \Delta G(\text{methylated}) - \Delta G(\text{unmodified})$. The uncertainty reflects the standard deviation from at least three measurements (see Supplementary Table 1)

destabilization of B0. This result shows that the destabilizing effects of m6A are not limited to bps in duplexes but can extend to bulges as well.

Similar results were obtained in the absence of $Mg^{2+}$ with the exception of HCV for which the destabilization was more significant in the absence ($1.4 \pm 0.4\ \text{kcal mol}^{-1}$) versus presence ($0.4 \pm 0.4\ \text{kcal mol}^{-1}$) of $Mg^{2+}$ (Fig. 2c). Together, these studies indicate that the observed m6A dependent stabilization requires a specific motif in which an m6A–U bp is neighbored by a 5′ guanine bulge.

**Abundance of junctional m6A base pairs transcriptome-wide.** Prior transcriptome-wide nuclease mapping studies of methylated RNA from human cells revealed a clear transition in the average RNA structure at the DRACH motif upon methylation[28]. In the methylated RNA, the purine nucleotide immediately 5′ to m6A had a much higher probability of being single-stranded whereas the methylated adenine had a slightly higher probability of being double-stranded[28]. The two nucleotides immediately 3′ to m6A showed a strong tendency to be paired. Similar results were observed with in vitro and in vivo SHAPE experiments[35] although the m6A site showed slightly higher reactivity, indicating a greater tendency to be unpaired. These results indicate that methylation thermodynamically biases m6A in cellular mRNAs to be located between single-stranded unpaired RNA and adjacent to helices[28].

Remarkably, the m6A stabilized B5′ motif identified in this work captures the unique conformational signatures induced by m6A in transcriptome-wide studies. The motif partially protects m6A by positioning it at a junction, exposes the 5′ guanine bulge, while the 3′ neighbors are helical. The abundance of the 5′B-[m6A–X] (where B is any bulge nucleotide and X is any nucleotide) motif could increase significantly on methylation because of two combined effects. On the one hand, m6A destabilizes bps in duplexes as well as bulges by as much as $\sim 1.7\ \text{kcal mol}^{-1}$. On the other hand, it can stabilize the 5′B-[m6A–U] motif by $\sim 1\ \text{kcal mol}^{-1}$. The combined contribution of $\sim 3\ \text{kcal mol}^{-1}$ could significantly bias the RNA folding landscape

away from structures in which m6A is embedded within duplexes and bulges toward 5′B-[m6A–U] and perhaps other 5′B-[m6A–X] motifs and to a degree that help account for the observed changes in transcriptome-wide structure-mapping data. Such a strong bias is required to explain the changes in RNA conformation induced by methylation especially since m6A occurs substoichiometrically in most RNAs[5,42].

To test this hypothesis, we asked what fraction of the 140,574 m6A sites that have been mapped throughout the human transcriptome[43] are predicted to fold into the 5′B-[m6A–U] or related 5′B-[m6A–X] motifs as the energetically most favorable secondary structure. We then asked what additional sites are predicted to fold into the 5′B-[m6A–U] or 5′B-[m6A–X] motifs as higher energy conformations that are within the $\sim 3\ \text{kcal mol}^{-1}$ energetic threshold. These conformations could be remodeled by m6A such to adopt the 5′B-[m6A–U] or 5′B-[m6A–X] motifs as the most energetically stable conformation. In particular, we used MC-Flashfold[39] to predict secondary structures for 41 nt fragments chosen such that the m6A residue was positioned in the middle. Predicted RNA structures were classified according to the secondary structural context and position of m6A (Fig. 3a). As a control, secondary structures were also predicted for 140,574 randomly selected RNA sequences from the same transcriptome that do not contain the m6A consensus sequence.

Approximately 4 and 22% of the m6A sites are predicted to fold into the 5′B-[m6A–U] or 5′B-[m6A–X] motif respectively as the most energetically favorable secondary structure (Fig. 3b). Similar abundances were obtained for the control unmodified sequences (Fig. 3b), indicating that sequences selected for methylation do not have an enhanced propensity to fold into the 5′B-[A–X] motif relative to random sequences. Similar results were also obtained when using a different RNA structure prediction program (RNAstructure[44]) and when varying the position of m6A or length of the sequences subjected to structure prediction (Fig. 3b and Supplementary Fig. 5a).

A much larger fraction of m6A sequences ($\sim 42$ and $\sim 68\%$ for 5′B-[m6A–U] and 5′B-[m6A–X], respectively) are predicted to fold into the junctional motif with energies $< 3\ \text{kcal mol}^{-1}$ relative

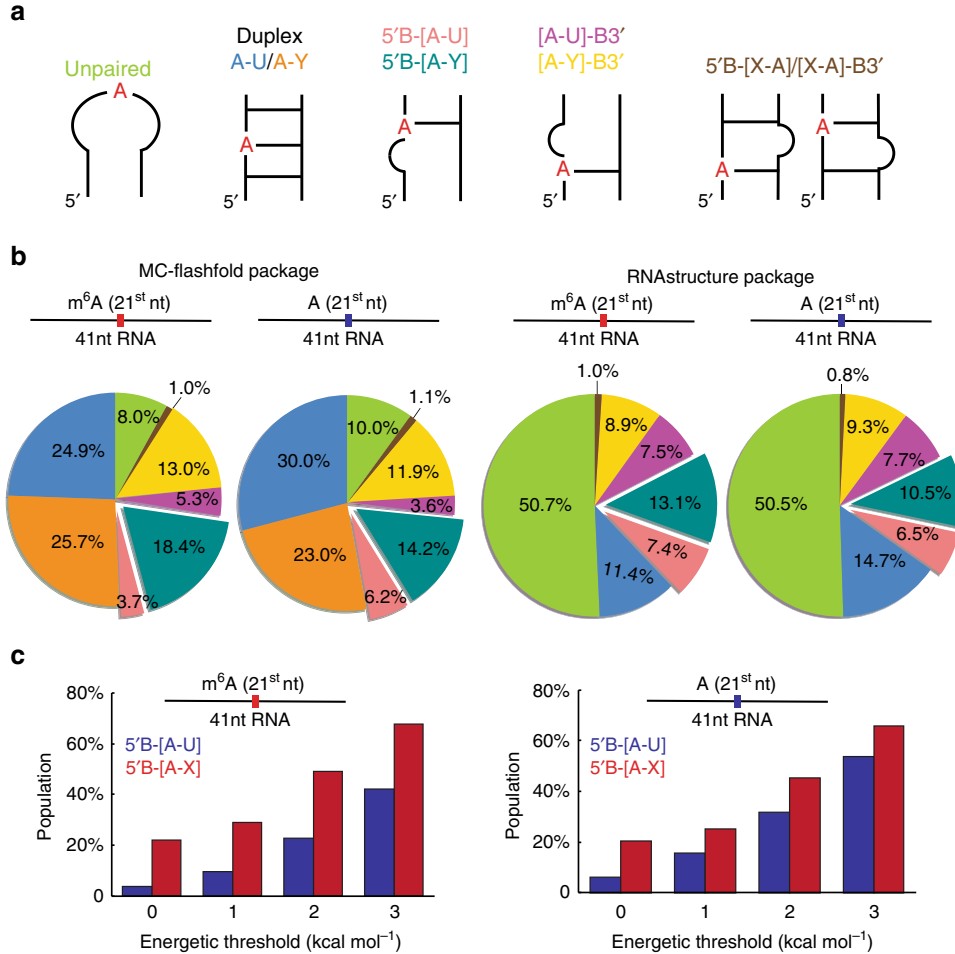

**Fig. 3** Predicted secondary structures of m⁶A containing RNA sequences. **a** Predicted RNA secondary structures were classified into the following motifs: Junctional Watson-Crick m⁶A–U or mismatch m⁶A–Y (Y denotes A or C or G.) with the bulge located 5′ (5′B-[A–U] and 5′B-[A–Y]) or 3′ ([A–U]-B3′ and [A–Y]-B3′) to m⁶A; or with the bulge located 5′ or 3′ (5′B-[X-A]/[X-A]-B3′, X denotes A or C or U or G) to m⁶A partner nucleotide. **b** Distribution of the lowest energy predicted structures for 140,574 m⁶A sites identified using individual-nucleotide-resolution cross-linking and immunoprecipitation (miCLIP) [43] in the human transcriptome (hg19) were predicted using RNAstructure[44] and MC-Flashfold[39]. Secondary structures were predicted for RNA sequences of 41-nt long with m⁶A (or A) in the middle. Also shown are corresponding predictions for 140,574 unmodified adenine sites selected randomly from the same human transcriptome. **c** Population of m⁶A and A sites predicted (MC-Flashfold) to fold into the 5′B-[A–U] and 5′B-[A–X] conformation within different energetic thresholds relative to the lowest energy structure

to the lowest energy structure (Fig. 3c). Similar results were obtained when predicting structures using RNAstructure[44] (Supplementary Fig. 5b). These higher energy structures could become the most stable structures upon methylation. While a similar increase in abundance is observed for the control unmodified sequences (Fig. 3c), indicating that the potential to form 5′B-[A–X] motif is not sequence dependent, these sequences are less likely to be methylated and to experience the energetic bias. Consequently only 20% of random A sites are expected to fold into 5′B-[A–X] motif as the minimum free energy structure. Therefore, while further experiments are needed to assess how m⁶A stabilizes / destabilizes various motifs in different sequence contexts, these results indicate that m⁶A could strongly bias RNA folding toward 5′B-[m⁶A–X] motifs, and thus help explain the m⁶A induced RNA conformational changes observed in transcriptome-wide structure mapping data.

**m⁶A is recognized in 5′G-[m⁶A–U] but not in duplexes**. To examine the functional significance of the 5′G-[m⁶A–U] motif, we examined whether it can be recognized by the YTH domain m⁶A reader (residues 380-579) from the human YTHDF2

(NP_057342.2) protein[45]. Binding was measured using a fluorescence polarization (FP) assay employing fluorescein tagged RNA[46]. Experiments were initially carried out in the absence of Mg²⁺ to allow comparison with previous studies[45,47–50].

As a positive control, the YTH domain binds to single-stranded RNA (SS) containing m⁶A and the consensus sequence (GGm⁶ACU) with $K_D = 0.18 \pm 0.02 \, \mu M$ (Fig. 4a) in good agreement with previously reported values (0.2-2 μM)[45,47–49]. The binding affinity is diminished at least ~ 100-fold ($K_D > 30 \, \mu M$) for unmodified SS (Fig. 4a), in good agreement with 10–100-fold weaker affinity reported previously[45,47–49].

Surprisingly, the YTH domain binds weakly ($K_D > 50 \, \mu M$) to both unmodified and methylated duplex RNA (D) (Fig. 4b). In both cases, the affinity is comparable to that measured for unmodified SS (Fig. 4b). This indicates that m⁶A sites in helical m⁶A–U Watson-Crick bps are not accessible for recognition by the YTH domain. This result is significant considering that helical A–U Watson-Crick bps occur frequently in the structures of RNA.

In contrast, the YTH domain does bind tightly to methylated B5′ (but not its unmodified counterpart; $K_D > 40 \, \mu M$) (Fig. 4c). The binding curve deviates from a simple two-state model and

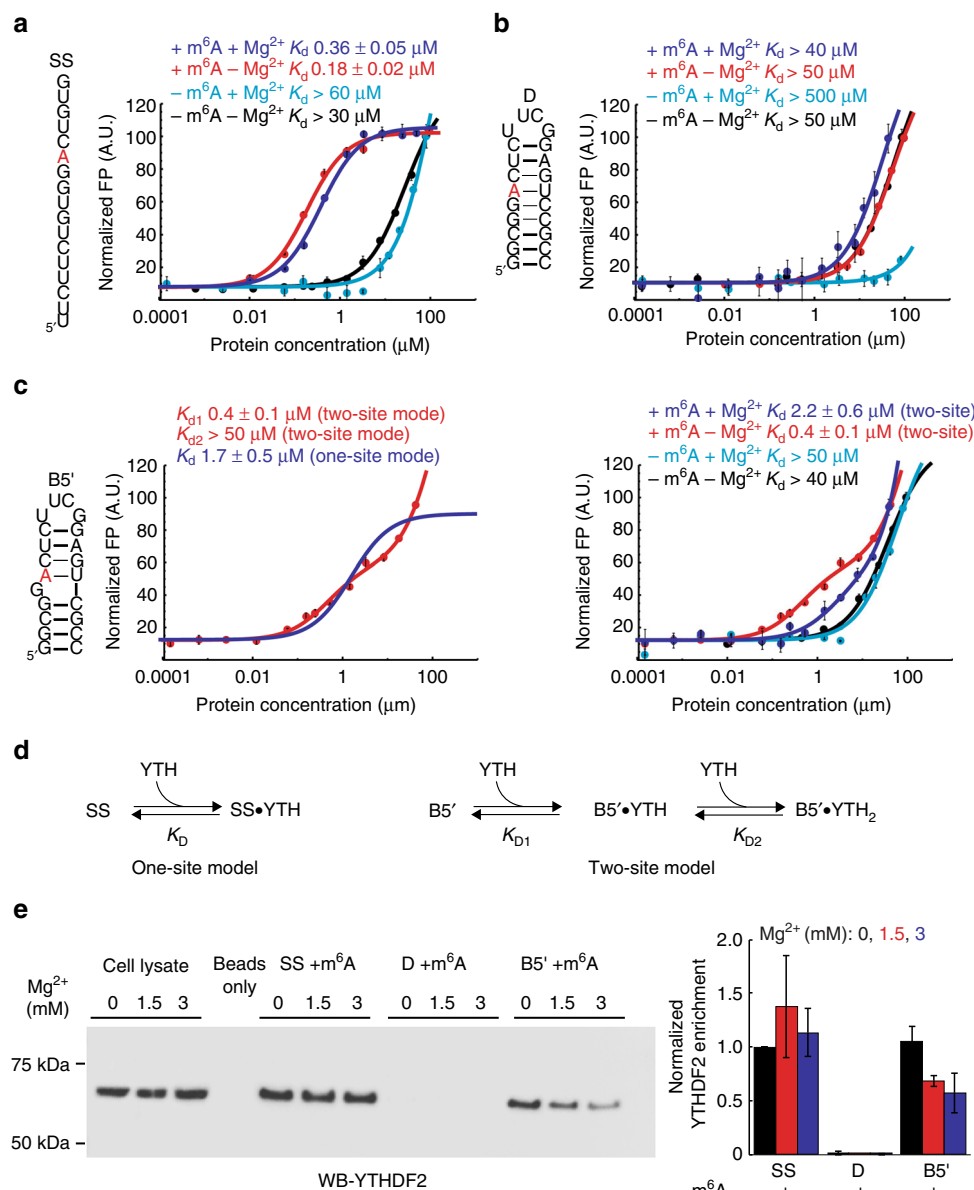

**Fig. 4** YTH recognizes m6A in B5′ but not D. **a–c** Binding curves for YTH domain with **a** SS, **b** D, and **c** B5′. **d** One-site and independent two-site binding models used for curve fitting. Data is shown for unmodified and methylated RNA with and without 3 mM Mg2+. Uncertainty reflects the standard deviation from three measurements. (**c**, left) Fits to the data assuming single (blue) versus two independent (red) binding sites (see Methods). **e** In vitro RNA pulldown assay using 5′biotinylated RNA constructs with m6A. Showing YTHDF2 binds preferentially to m6A in unpaired context or at the junction, and Mg2+ decreases the enrichment of YTHDF2 by methylated B5′. Blots shown are representative of results from three experiments. Uncropped blots are shown in Supplementary Fig. 6. The semi-quantitative YTHDF2 enrichment analysis was performed using ImageJ[56]. The uncertainty represents the standard deviation from three experiments

the data is better fit to a model that assumes two independent binding sites (Fig. 4d). A similar behavior was previously observed for YTH domain binding to methylated ssRNA[45]. One of two $K_D$s obtained from this analysis ($K_{D1} = 0.3 \pm 0.1 \mu M$) is consistent with high affinity binding as observed for methylated SS while the other ($K_{D2} > 50 \mu M$) is consistent with weak binding as observed with unmodified RNA. It is possible that the YTH domain specifically binds to m6A in B5′ with high affinity but also weakly and non-specifically binds to other parts of the RNA (e.g. duplex or apical loop region). Alternatively, the YTH domain could bind to m6A in two different RNA conformations (e.g. helical versus more single-stranded). These results indicate that in contrast to duplexes, m6A in 5′B-[m6A–U] is recognized by the YTH domain.

Based on the NMR data, Mg2+ stabilizes the junctional m6A–U bp in B5′ potentially making m6A6 less accessible for recognition. Indeed, the binding affinity to methylated B5′ decreased by ~6-fold ($K_{D1}$ $0.4 \pm 0.1 \mu M$ versus $2.2 \pm 0.6 \mu M$, Fig. 4c) in the presence of 3 mM Mg2+ but remained ~10 fold tighter than that of unmodified SS or B5′ and ~20-fold tighter than methylated D. By comparison, the binding affinity of YTH domain to methylated or unmodified SS and D as well as unmodified B5′ was only slightly (by ~1-fold) weakened in the presence of Mg2+ (Fig. 4a, b).

We confirmed the above results for the full-length YTHDF2 protein using in vitro RNA pulldown experiments (Fig. 4e). Here, biotinylated RNA baits were incubated with HEK293T cell lysates, and RNA:protein complexes were purified with

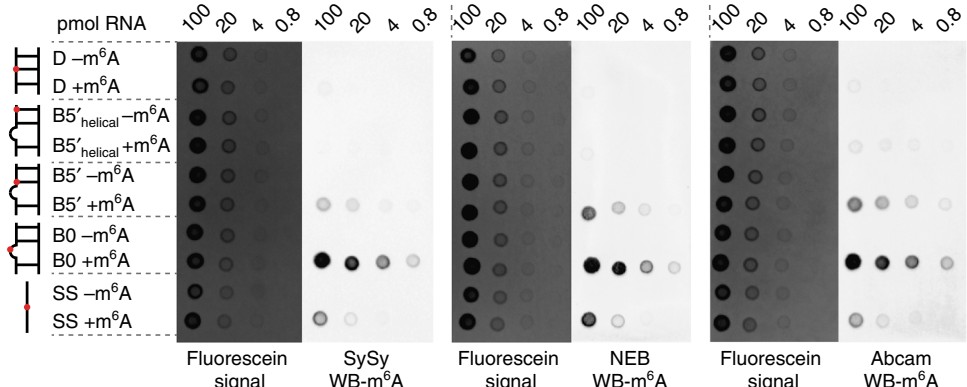

**Fig. 5** m6A antibodies recognize m6A in B5′ but not D. Dot blot assays assessing the antibody specificity for m6A in different secondary structure contexts. Increasing amounts of 5′ fluorescein labeled RNAs with and without m6A were spotted onto the membrane. The quantity of RNA was visualized by the fluorescein signal. Three different m6A antibodies (SySy polyclonal, NEB monoclonal, Abcam polyclonal) were tested. Red dots in the schematic secondary structures represent the m6A sites. The SS construct used in dot blot assays is 5′-GGCGGm6ACUC-3′. Blots shown are representative of results from three experiments

streptavidin agarose. The bound proteins were then eluted and subjected to SDS-PAGE, followed by western blot to detect YTHDF2. In agreement with results from the FP assay, in the absence of $Mg^{2+}$, modified B5′ enriches YTHDF2 protein by an amount comparable to modified SS. While similar enrichments were observed in the absence and presence of $Mg^{2+}$ for SS, the enrichment with B5′ decreased 1-fold in the presence of $Mg^{2+}$. In sharp contrast, YTHDF2 was barely enriched by methylated D in the absence and presence of $Mg^{2+}$.

Similar results were also obtained when examining how well m6A is recognized by three m6A antibodies (Abcam polyclonal antibody, SySy polyclonal antibody and NEB monoclonal antibody) that are used in transcriptome-wide studies. As expected, all unmodified RNA constructs were not well recognized by the antibodies in dot blot assays (Fig. 5). Both methylated SS and B0 exhibited strong signals, suggesting that the antibodies can recognize unpaired m6A (Fig. 5 and Supplementary Fig. 5c). Once again, the methylated D and methylated B5′helical were poorly recognized by all three antibodies. This was also the case for a 9-mer duplex lacking the stabilizing UUCG apical loop (Supplementary Fig. 5c). In contrast, the junctional m6A–U bp in B5′ was well recognized by all three antibodies.

Taken together, these results indicate the m6A within the 5′G-[m6A–U] motif is well recognized by YTH reader proteins as well as commonly used m6A antibodies, whereas m6A–U bps embedded in duplexes are not.

## Discussion

A growing number of studies indicate that m6A can exert biological effects by modulating RNA structure. So far, studies have primarily focused on the destabilization of base pairing due to steric collisions with the methyl group[28,29,31,33]. However, m6A has also been shown to stabilize duplexes when placed at dangling ends and within apical loops[28,29]. Our results identify a 5′ bulge motif that is predicted to be abundant and stabilized by m6A in a $Mg^{2+}$ dependent manner. The 5′ bulge most likely provides m6A the conformational freedom needed to allow the methyl group to optimally stack with the 3′ neighbor while maintaining Watson-Crick m6A–U pairing. $Mg^{2+}$ may help to stabilize the resulting backbone conformation at the bulge by neutralizing electrostatic repulsion. In this regard, it is noteworthy that $Mg^{2+}$ preferentially binds to guanine and adenine nucleotides[51], thus the naturally occurring DRACH motif may enhance the propensity for the 5′ neighbor to bulge out. This motif with base-paired m6A stacking

on its 3′ neighbor and with a 5′-neighboring bulged nucleotide can potentially form in a variety of contexts, including bulges, apical loops, and possibly other higher order structures. Further studies are needed to assess the robustness of the observed stabilization across these different structural and sequence contexts.

While biophysical studies show that m6A destabilizes pairing of adenine nucleotides in vitro, these results were at odds with in vitro transcriptome-wide nuclease mapping[28] and SHAPE[35] data showing that methylation primarily acts to increase the single-stranded character of the purine nucleotide immediately 5′ to the m6A site, while the adenine nucleotide itself was less affected[28]. These data were obtained devoid of cellular proteins and the conformational differences likely originate directly from the impact of m6A on RNA structure. A bias toward the junctional m6A bps flanked by a 5′ bulge following methylation could help explain the increase in single-stranded character observed specifically for the purine nucleotide 5′ to the m6A bulge as well as the increase in the double-stranded character observed for the m6A nucleotide using transcriptome-wide nuclease mapping experiments[28]. This bias could be significant especially when considering that m6A selectively stabilizes the 5′B-[A–U] motif but destabilizes other common motifs such as duplexes and bulges. It should be noted, however, that in vivo SHAPE data[35] suggest that the impact of methylation on RNA structure can be more complex, possibly due to interactions with proteins.

Our results show that m6A is effectively recognized by both YTH domain reader proteins and m6A antibodies when located in the 5′G-[A–U] motif. The motif could therefore also serve functional roles recruiting YTH domains and is likely to be identified in current methods used to map m6A transcriptome-wide. In contrast, our results indicate that both YTH domain reader proteins and m6A antibodies do not recognize m6A–U bps embedded within duplexes. Current methods for m6A mapping attempt to remove these structural biases through nuclease digestion and heating of the RNA prior to immunoprecipitation. However, the RNA fragments are still 30–130 nt long[43] and are subjected to cooling on ice following heating, allowing for re-annealing of the RNA. Indeed, we found that m6A sites buried in A–U bps within duplexes were poorly recognized by the antibodies even when increasing the temperature and/or duration of heating with or without 8 M urea. This raises the possibility that there are m6A sites buried within duplexes that have evaded detection using current transcriptome-wide methods. This could potentially confound the interpretation of m6A induced conformational changes based on transcriptome-wide structure

mapping data since motifs in which m⁶A is accessible such as 5′G-[A–U] could be over-represented compared with those in which m⁶A is not accessible. Additional studies are needed to examine whether these structure-specific biases in m⁶A recognition also occur during immunoprecipitation experiments with free-floating RNA used in the transcriptome-wide mapping of m⁶A sites.

Our results indicate that while m⁶A destabilizes duplex A–U bps and disrupts base stacking in certain motifs, the structure landscape of the epitranscriptome is also likely punctuated by motifs such as 5′G-[m⁶A–U] that are stabilized by m⁶A. This motif presents a readable m⁶A site in a structured region of RNA. While the biological significance of this motif, including whether it is recognized by specific RNA-binding proteins, requires further investigation, our findings expand the mechanisms by which m⁶A can modulate RNA structure and should facilitate the interpretation of transcriptome-wide data.

## Methods

**Sample preparation.** Modified ($N^6$-methylated adenosine, 5′-biotinylated RNA and 5′-fluorescein labeled RNA) and unmodified RNA oligonucleotides were synthesized using a MerMade 6 Oligo Synthesizer employing 2′-tBDSilyl protected phosphoramidite (ChemGenes) on 1 μmol standard synthesis columns (1000 Å) (BioAutomation). n-acetyl protected rC, rA, and rG phosphoramidites were used to avoid incomplete deprotection when using isobutyryl protected rG and benzoyl protected rA phosphoramidites. m⁶A phosphoramidite was purchased from ChemeGenes, biotin phosphoramdites, and fluorescein phosphoramdites were purchased from Glen Research. The $^{15}$N1-labeled guanosine and the $^{15}$N3-labeled uridine phosphoramidites were synthesized according to a published procedure[52]. ssRNA oligonucleotides were synthesized with the option to leave the final 5′-protecting group (4,4′-dimethoxytrityl (DMT)) on for 2′O deprotection and cartridge purification. UUCG-capped hairpin RNAs were synthesized with DMT group off for DMT-off 2′O deprotection and PAGE purification, since regular 2′O deprotection method might cause incomplete deprotection for UUCG-capped samples. Synthesized oligonucleotides were cleaved from the 1 μmol column using 1 mL ammonia methylamine (1:1 ratio of 30% ammonium hydroxide and 30% methylamine) followed by 2-hour incubation at room temperature to allow base deprotection. The solution was then air-dried and dissolved in 115 μL DMSO, 60 μL TEA, and 75 uL TEA-3HF for regular 2′O deprotection, or in 100 μL DMSO and 125 μL TEA-3HF for DMT-off 2′O deprotection, followed by 2.5 h incubation at 65 °C. Regular 2′O deprotected samples were then quenched with Glen-Pak RNA quenching buffer and loaded onto Glen-Pak RNA cartridges (Glen Research Corporation) for purification using the online protocol (http://www.glenresearch.com/). Samples were then ethanol precipitated, air dried, dissolved in water and buffer exchanged or diluted into the desired buffer (15 mM sodium phosphate, 25 mM NaCl, 0.1 mM EDTA, 10% D₂O, pH 6.4 with or without 3 mM Mg²⁺). DMT-off 2′O deprotected samples were directly ethanol precipitated, and purified using 20% (w/v) denaturing PAGE, and electroeluted into 20 mM Tris buffer, pH 8, and subsequently ethanol precipitated. The samples were then dissolved in water (50 μM for hairpin and 200–500 μM for duplex), annealed by heating at 95 °C for 10 min, and cooled on ice for 30 min (for hairpin) or at room temperature for 2 h (for duplex). RNA samples were buffer exchanged at least three times using a centrifugal concentrator (EMD Millipore) into the desired buffer. The purity of methylated and unmodified oligonucleotides was verified using NMR to monitor resonances from deprotecting groups and for B5′ using liquid chromatography-mass spectrometry LC/MS (Novatia).

**NMR spectroscopy.** All NMR experiments were collected on a 600 MHz Bruker NMR spectrometer equipped with an HCN cryogenic probe. Data were processed and analyzed using NMRpipe[53] and SPARKY (T.D. Goddard and D.G. Kneller, SPARKY 3, University of California, San Francisco), respectively. Resonances were assigned using 2D HSQC, HMQC, ¹H-¹H NOESY (mixing times of 100 ms and 150 ms), and HCN experiments in the absence of Mg²⁺.

**UV melting.** Thermal melting experiments were conducted on a PerkinElmer Lambda 25 UV/VIS spectrometer with a RTP 6 Peltier Temperature Programmer and a PCB 1500 Water Peltier System. All RNA samples were buffer exchanged at least three times with a centrifugal concentrator (EMD Millipore) to desired buffers (15 mM sodium phosphate, 25 mM NaCl, 0.1 mM EDTA, pH 6.4 with or without 3 mM Mg²⁺), followed by direct dilution to 3 μM with the same buffer. At least three measurements were carried out for each RNA with a sample volume of 400 μL in a Teflon-stoppered 1 cm path length quartz cell. The absorbance at 260 nm was monitored while the temperature was varied between 15 and 95 °C.

Thermodynamic parameters from UV melting experiments were fitted using nonlinear model fitting in Mathematica 10.0 (Wolfram Research) such that melting

temperature ($T_m$) and enthalpy ($\Delta H$) for duplex and hairpin association were obtained by the fitting to Eqs. (1) and (2), respectively[54],

$$f_{\text{duplex}} = \frac{1 + 4e^{(\frac{1}{T_m} - \frac{1}{T})\frac{\Delta H}{R}} - \sqrt{1 + 8e^{(\frac{1}{T_m} - \frac{1}{T})\frac{\Delta H}{R}}}}{4e^{(\frac{1}{T_m} - \frac{1}{T})\frac{\Delta H}{R}}} \quad (1)$$

$$f_{\text{hairpin}} = \frac{e^{(\frac{1}{T_m} - \frac{1}{T})\frac{\Delta H}{R}}}{1 + e^{(\frac{1}{T_m} - \frac{1}{T})\frac{\Delta H}{R}}} \quad (2)$$

where $T$ is the temperature (K), $R$ is the gas constant (kcal mol⁻¹), and $f$ is the fraction of folded duplex or hairpin for Eqs. (1) and (2), respectively. $\Delta G$ and $\Delta S$ were calculated from Eqs. (3) and (4) for duplex association and hairpin folding respectively.

$$\Delta S = \frac{\Delta H}{T_m} - R\ln(\frac{C_T}{2}); \; \Delta G = \Delta H - T\Delta S \quad (3)$$

$$\Delta S = \frac{\Delta H}{T_m}; \; \Delta G = \Delta H - T\Delta S \quad (4)$$

where $C_T$ is the total concentration of RNA. The uncertainty in $T_m$, $\Delta H$, $\Delta G$, and $\Delta S$ was obtained based on the standard deviation in triplicate measurements. The destabilization or stabilization effects of m⁶A was calculated using the following equations:

$$\Delta\Delta G = \Delta G_{\text{m6A}} - \Delta G_{\text{unmodified}} \quad (5)$$

$$\Delta\Delta H = \Delta H_{\text{m6A}} - \Delta H_{\text{unmodified}} \quad (6)$$

$$\Delta\Delta S = \Delta S_{\text{m6A}} - \Delta S_{\text{unmodified}} \quad (7)$$

**Secondary structure prediction.** The programs MC-Flashfold package[39] and RNAstructure package[44] were used to predict the secondary structure of 140,574 sequences containing m⁶A that were identified using single-nucleotide transcriptome-wide m⁶A mapping of human genome build (hg19)[43]. Another 140,574 control sequences that do not contain the DRACH motif and that were randomly selected from the same transcriptome were also analyzed. In-house Python scripts were used to perform the analysis. The sequences were 41 nt long with m⁶A positioned in the middle. Analysis was repeated when varying the position and length of the m⁶A site (31-nt long sequences with m⁶A site at the eleventh or twentyfirst position). The lowest energy predicted structures were classified into different categories: duplex, unpaired, 5′B-[A–U], 5′B-[A–Y] (Y denotes A or C or G), [A–U]-B3′, [A–Y]-B3′ and 5′B-[X–A]/[X–A]-B3′ (X denotes A or C or G or U). 5′-B-[A–U] and 5′B-[A–Y] are motifs containing junctional A–U or A–Y bp with unpaired residues at 5′ side of A. Similarly, [A–U]-B3′, [A–Y]-B3′ are motifs containing junctional A–U or A–Y bps with unpaired residues at 3′ side of A. 5′B-[X–A]/[X–A]-B3′ is the motif with junctional A–X bps that are on the opposite strand of unpaired residues. With the exception of G–U bps, the RNAstructure package does not predict mismatches. To calculate the population of 5′B-[A–Y] and [A–Y]-B3′, any A–Y site that was immediately adjacent to a canonical Watson–Crick bp was counted as a mismatch. The MC-Flashfold package does predict mismatches allowing further classification of the duplex category into duplex (A–U) and duplex (A–Y). MC-Flashfold was used to predict the secondary structures setting the maximum energy difference of output structures to minimum free energy (MFE) of 0, 1, 2 or 3 kcal mol⁻¹. For RNAstrutcure package, the maximum percent energy difference between predicted structures and MFE was set to 10, 20, or 30%. The population of 5′B-[A–U] or 5′B-[A–X] was determined to be the number of sequences predicted to form 5′B-[A–U] or 5′B-[A–X] in the output structures divided by the total number of sequences (140,574).

**Protein expression and purification.** A gene, codon optimized (Supplementary Table 2) for *Escherichia coli* expression, encoding the YTH domain (residues 380–579) of human YTHDF2 (NP_057342.2) was subcloned into the pET15b vector (the plasmid containing the codon optimized sequence was purchased from GenScript). The plasmid was transformed into *E. coli* strain C41(DE3). The cells were grown in LB medium containing 50 μg ml⁻¹ ampicillin at 37 °C to an OD₆₀₀ between 0.4 and 0.6. Recombinant proteins were induced by adding isopropyl-β-D-thiogalactopyranoside (IPTG) to 0.5 mM and overexpressed in LB medium at 20 °C overnight. Cells expressing YTH domain were harvested at 4 °C and stored at −80 °C or lysed immediately using a microfluidizer in the buffer containing 20 mM HEPES (pH 7.4), 200 mM NaCl, 1 mM DTT supplemented with protease inhibitors and DNase. Following centrifugation at 17,500 rpm for 3 min at 4 °C, the supernatant was loaded to Ni-NTA affinity column. The eluted protein was further purified by size exchange chromatography (SEC) using a Superdex 75 pg column. The peak fractions were collected and concentrated. The protein purity was

assessed using SDS-PAGE analysis and protein concentration determined using Bradford assays.

**YTH domain binding assays**. Binding experiments were carried out using a previously reported[46] fluorescence polarization (FP) assay at 25 °C and using a PanVera Beacon 2000 instrument (Invitrogen, Madison, WI, USA). 5′-fluorescein-labeled RNAs were synthesized and purified as described in the sample preparation section. Fluorescein labeled samples were dissolved in water and then directly diluted into the desired buffer (20 mM HEPES 50 mM NaCl, 3 mM DTT, pH 8.2, with or without 3 mM Mg$^{2+}$). YTH domain protein was serially diluted into 200 µL of the same binding buffer containing 2 nM 5′-fluorescein-labeled RNA. Fluorescence polarization was measured at an excitation wavelength of 490 nm and an emission wavelength of 530 nm. The total fluorescent intensities did not vary significantly throughout the measurements (i.e., $(I_{max} - I_{min})/I_{min}$ ~5%, in which $I_{max}$ and $I_{min}$ are the maximum and minimum intensities, respectively). The binding curves were fitted to either one-site (Eq. (8)) or two-site[55] (Eq. (9)) binding mode using Mathematica 10.0 (Wolfram Research).

$$A_t = \frac{A - (A - B) \times (R_t + L_t + K_D - \sqrt{(R_t + L_t + K_D)^2 - 4 \times R_t \times L_t})}{2 \times R_t} \quad (8)$$

$$A_t = \frac{A + \frac{B \times L}{K_{D1}} + \frac{C \times L^2}{K_{D1} \times K_{D2}}}{1 + \frac{L}{K_{D1}} + \frac{L^2}{K_{D1} \times K_{D2}}} \quad (9)$$

$L_t$ is the total concentration of protein, $R_t$ is the concentration of RNA. $A$ and $B$ represent free RNA anisotropy and RNA-protein intrinsic anisotropy of the saturated protein-RNA complex. $K_D$ is the dissociation constant. In Eq. (9), $K_{D1}$ and $K_{D2}$ are dissociation constants of the two binding events respectively. $A$, $B$, and $C$ represent free RNA anisotropy, RNA-Protein and RNA-[Protein]$_2$ intrinsic anisotropy of the saturated protein-RNA complexes. By titrating excess protein against 2 nM RNA, $L_{free}$ remains in excess over concentrations of RNA-Protein and RNA-[Protein]$_2$ complexes. $L_{free}$ is therefore well approximated by $L_t$ and is referred to simply as $L$.

**In vitro RNA pulldown**. Biotinylated RNA samples were synthesized and purified as described in the sample preparation section. HEK293T cells were harvested at ~70–80% confluency, washed with cold PBS and lysed by dounce homogenization in lysis buffer (10 mM NaCl, 2 mM EDTA, 0.5% Triton X-100, 0.5 mM DTT, 10 mM Tris, pH 7.4), and the lysate was then brought to 150 mM KCl and 5% Glycerol (v/v). Mammalian protease inhibitor cocktail (Sigma-Aldrich) and phosphatase inhibitor cocktail (Sigma-Aldrich) were freshly added to the lysis buffer. The following steps were performed as described previously[5], except that binding buffers containing different Mg$^{2+}$ concentrations were used (10 mM Tris pH 7.5, 150 mM KCl, 0.5 mM DTT, 0.05% (v/v) NP-40 and 0 mM, 1.5 mM, or 3 mM MgCl$_2$). Proteins were eluted under mild conditions with elution buffer (50 mM Tris, pH 7.5, 200 mM NaCl, 2% SDS (w/v), and 1 mM biotin) at 60 °C, 1100 rpm on a thermal block shaker. The pulldown eluent was loaded on 4–12% polyacrylamide Bis-Tris polyacrylamide gels (Thermofisher) and then transferred onto PVDF membranes (GE Healthcare, Amersham) using a wet electrophoretic transfer system (BioRad). The membrane was then blocked with 5% nonfat dry milk in 0.1% PBST (0.1% Tween-20 in 1× PBS, pH 7.4). Rabbit polyclonal YTHDF2 antibody (Aviva system biology ARP67917_P050) was diluted 1:1000 in 0.1% PBST and incubated on the membrane for 1 h at room temperature (15–25 °C) or overnight at 4 °C. Membranes were then washed in 0.1% PBST, and then incubated with HRP-conjugated goat anti-rabbit IgG (Abcam ab6721) at 1:2500 dilution in 0.1% PBST for 1 h at 25 °C. Membranes were then subsequently washed with 0.1% PBST and exposed with enhanced chemiluminescence (ECL; GE Healthcare).

**Dot blot assays**. RNA samples were quantified using UV spectroscopy. Methylated and unmodified RNA samples were spotted onto a nylon membrane (GE healthcare). Fluorescein labeled RNA samples (prepared as described in sample preparation section) were scanned by UVP imaging system to visualize the quantity of RNA spotted onto the membrane. The membrane was then UV-crosslinked and blocked for 1 h in 5% nonfat dry milk in 0.1% PBST (0.1% Tween-20 in 1× PBS, pH 7.4). The following steps were performed essentially as described for RNA pulldown (above). m$^6$A antibody (SySy 202003, NEB E1610S, or Abcam ab151230) was diluted 1:1000 in 0.1% PBST and incubated on the membrane for 1 h at room temperature or overnight at 4 °C. Membranes were then washed in 0.1% PBST, incubated with HRP-conjugated goat anti-rabbit IgG (Abcam ab6721) (1:2500) in 0.1% PBST for 1 h at room temperature. The membrane was then washed again in 0.1% PBST, and developed with enhanced chemiluminescence (ECL; GE Healthcare). The **SS** construct used in dot blot assay is 5′-GGCGGm$^6$ACUC-3′.

**Code availability**. The in-house Python scripts used for secondary structure prediction are available on request to the corresponding author.

**Data availability**. The authors declare that the data supporting the findings of this study are available within the article and its Supplementary Information files, or are available upon reasonable requests to the authors.

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

## Acknowledgements

We thank Nicole Orlovsky, Atul Rangadurai, and Mary Clay, members of the Al-Hashimi laboratory, and Dr. Tony Mustoe (UNC-Chapel Hill) for assistance and critical comments on the manuscript. We thank Dr. Richard Brennan for providing the UV-Vis spectrophotometer. We thank New England Biolabs (NEB) for providing the monoclonal m6A antibody as a gift to S.M.H. We acknowledge technical support and resources from the Duke Magnetic Resonance Spectroscopy Center. This work was supported by the US National Institutes of Health (P01GM0066275 to H.M.A., R01AI125416 to S.M.H. and R00MH104712 to K.D.M.) and Burroughs Welcome Fund (to S.M.H.). The research reported in this article was performed by the Duke University faculty, research associate and was funded by US National Institute of Health contract to H.M.A.

## Author contributions

B.L., D.K.M. and H.M.A. conceived the project and experimental design. B.L. and D.K.M. prepared NMR samples and performed NMR experiments and analyzed NMR data. R.P. and C.K. prepared $^{15}$N-N1/3 labeled Guanosine and Uridine phosphoramidite. B.L. performed UV melting experiments, transcriptome-wide RNA secondary structure prediction, and dot blot assays. B.L. performed the YTH domain protein binding assays, with assistance from M.A.S. B.L. and S.H.C performed the in vitro RNA pulldown experiments. H.M.A. and B.L. wrote the manuscript with critical input from D.K.M., K. D.M., M.A.S., S.M.H., and S.H.C.

## Additional information

**Competing interests:** H.M.A. is an advisor to and holds an ownership interest in Nymirum, an RNA-based drug discovery company. The remaining authors declare no competing interests.

