## [Peer Review File · Nature Communications]

Reviewers' comments:

Reviewer #1 (Remarks to the Author):

In this manuscript Liu et al. report a detailed investigation of the methyladenosine modification on RNA stability using a combination of biophysical techniques and translate their findings to transcriptome-wide data. The authors discuss their approach in the light of transcriptome-wide data and how they can improve the analysis of these data. The manuscript is very exciting, technically sound, a piece of novel work of high quality and should be considered for publication.

Minor points:

- Please add the number of experiments carried out in Figure 1 e, 2c.

Reviewer #2 (Remarks to the Author):

This manuscript presents a novel discovery of a stabilization effect of m6A in a secondary structure motif B5' in the presence of Mg²⁺, while m6A shows destabilization or none effect when located in the stem regions away from the bulge, opposite to the bulge or in the double-stranded RNA. The authors also report that this m6A-preferred structural motif (in the presence of Mg²⁺) can be readily recognized by the YTH domain reader proteins as well as m6A-specific antibodies. However, m6A embedded in double-stranded RNA cannot be recognized. This discovery of complex stabilization and destabilization effects of m6A reveals a more complicated, structure-dependent thermodynamics landscape, which can help explain biological impacts of the sequence contexts beyond the conserved sequence, i.e. the secondary structure near the m6A site. The fact that the antibody fails to recognize m6A in double-stranded RNA structures may indeed, as the authors suggest, lead to underestimation of highly structured m6A-containing transcripts. This observation urges the development of antibody-independent m6A mapping methods. This is a nice work. I do have a few comments:

- 1, In the discussion of the stabilization effect of m6A, regarding the proposed mechanism of stacking interaction between the methyl group and 5'-buldge, I wonder if this can be reflected in the enthalpic and entropic changes measured by the UV melting experiments.
- 2, It is not straightforward to understand line 195, "the m6A stabilized B5' motif identified in this work captures the unique conformational signatures induced by m6A in transcriptome-wide studies". It seems that though there is an increasing population of m6A-involved sequences identified within 3kcal/mol energy threshold that adopt 5'-B-[A-X] structure from 22% to 60%, this population is as big as the population of the random A site control that also presents 60% within 3 kcal/mol (Figure 3C). This point may need more explanation.
- 3, In the FP binding experiments, while the authors presented different models for fitting the binding curve between YTHDF2 and B5' construct. It would be nice if different binding models are also presented. In an FP experiment, it is always nice to make sure that the total of parallel and perpendicular intensities remain constant throughout the measurement to rule out other effects in the experimental setups.

Reviewer 1

In this manuscript Liu et al. report a detailed investigation of the methyladenosine modification on RNA stability using a combination of biophysical techniques and translate their findings to transcriptome-wide data. The authors discuss their approach in the light of transcriptome-wide data and how they can improve the analysis of these data. The manuscript is very exciting, technically sound, a piece of novel work of high quality and should be considered for publication.

We thank the reviewer for his/her positive comments.

Minor points:

- *Please add the number of experiments carried out in Figure 1 e, 2c.*

The exact number of experiments performed is now included in Supplementary Table 1. We now reference Supplementary Table 1 in the legends of Fig. 1 and Fig. 2 on pages 34 and 35, respectively.

Reviewer 2

This manuscript presents a novel discovery of a stabilization effect of m6A in a secondary structure motif B5' in the presence of Mg²⁺, while m6A shows destabilization or none effect when located in the stem regions away from the bulge, opposite to the bulge or in the double-stranded RNA. The authors also report that this m6A-preferred structural motif (in the presence of Mg²⁺) can be readily recognized by the YTH domain reader proteins as well as m6A-specific antibodies. However, m6A embedded in double-stranded RNA cannot be recognized. This discovery of complex stabilization and destabilization effects of m6A reveals a more complicated, structure-dependent thermodynamics landscape, which can help explain biological impacts of the sequence contexts beyond the conserved sequence, i.e. the secondary structure near the m6A site. The fact that the antibody fails to recognize m6A in double-stranded RNA structures may indeed, as the authors suggest, lead to underestimation of highly structured m6A-containing transcripts. This observation urges the development of antibody-independent m6A mapping methods. This is a nice work. I do have a few comments:

We thank the reviewer for his/her positive comments.

1, In the discussion of the stabilization effect of m6A, regarding the proposed mechanism of stacking interaction between the methyl group and 5'-bulge, I wonder if this can be reflected in the enthalpic and entropic changes measured by the UV melting experiments.

We thank the reviewer for this excellent suggestion. The UV melting results suggest that the stabilization effect of m⁶A on B5' is indeed driven by more favorable enthalpy and less favorable entropy (Supplementary Table 1), which is consistent with more favorable interactions with m⁶A. To address this comment, we now include the following sentence on page 7 in the revised manuscript:

“Based on the thermodynamic parameters obtained from the UV experiments (Supplementary Table 1), the observed m⁶A-mediated stabilization of B5' is driven by more favorable enthalpy, which is consistent with formation of favorable structural interactions.”

2, It is not straightforward to understand line 195, “the m6A stabilized B5' motif identified in this work captures the unique conformational signatures induced by m6A in transcriptome-wide studies”. It seems that though there is an increasing population of m6A-involved sequences identified within 3kcal/mol energy threshold that adopt 5'-B-[A-X] structure from 22% to 60%, this population is as big as the population of the random A site control that also presents 60% within 3 kcal/mol (Figure 3C). This point may need more explanation.

We clarified this point on page 11:

“While a similar increase in abundance is observed for the control unmodified sequences (Fig. 3c), indicating that the potential to form 5'B-[A-X] motif is not sequence dependent, these sequences are less likely to be methylated and to experience the energetic bias. Consequently only 20% of random A sites are expected to fold into 5'B-[A-X] motif as the minimum free energy structure.”

3, In the FP binding experiments, while the authors presented different models for fitting the binding curve between YTHDF2 and B5' construct. It would be nice if different binding models are also presented. In an FP experiment, it is always nice to make sure that the total of parallel and perpendicular intensities remain constant throughout the measurement to rule out other effects in the experimental setups.

We now show the two binding models used to fit the data in Fig. 4d. We also have included the following sentence regarding the FP experiments the total intensities throughout the measurement did not vary significantly (i.e. $(I_{\max}-I_{\min})/I_{\min}$ on page 27:

“The total fluorescent intensities did not vary significantly throughout the measurements (i.e. $(I_{\max}-I_{\min})/I_{\min} \sim 5\%$, in which I_{\max} and I_{\min} are the maximum and minimum intensities respectively).”